# Orogenital Human Papillomavirus Infection and Vaccines: A Survey of High- and Low-Risk Genotypes Not Included in Vaccines

**DOI:** 10.3390/vaccines11091466

**Published:** 2023-09-07

**Authors:** Michela Buttà, Nicola Serra, Vera Panzarella, Teresa Maria Assunta Fasciana, Giuseppina Campisi, Giuseppina Capra

**Affiliations:** 1Department of Health Promotion, Mother and Child Care, Internal Medicine and Medical Specialties (ProMISE) “G. D’Alessandro”, University of Palermo, 90133 Palermo, Italy; michela.butta@unipa.it (M.B.); teresa.fasciana@unipa.it (T.M.A.F.); 2Department of Surgical, Oncological and Oral Sciences (Dichirons), University of Palermo, 90133 Palermo, Italy; vera.panzarella@unipa.it (V.P.); giuseppina.campisi@unipa.it (G.C.); 3Department of Public Health, University Federico II of Naples, 80138 Napoli, Italy; nicola.serra@unina.it; 4Microbiology and Virology Unit, Polyclinic Hospital “P. Giaccone”, 90133 Palermo, Italy; 5Oral Medicine with Dentistry for Fragile Patients Unit, Polyclinic Hospital “P. Giaccone”, 90133 Palermo, Italy

**Keywords:** human papillomavirus, HPV, HPV vaccines, oral infection, orogenital infection, high- and low-risk genotypes

## Abstract

Knowledge of human papillomavirus transmission from the genital tract to the oral mucosa remains unsatisfactory, with poor and often inconsistent literature results. The increase in HPV-associated oral malignancies prompts further analysis of the simultaneous detection of the virus in the two anatomical areas and on the identification of genotypes to be included in future vaccines. Therefore, in this retrospective study, we evaluated orogenital HPV concurrence, hrHPV, lrHPV and type-concordance in 337 samples, as well as the prevalence of the most common genotypes not included in HPV vaccines. Concurrence was found in 12.5% (31/248) of cases, hr-concordance in 61.3% (19/31) and lr-concordance in 12.9% (4/31). Finally, type-concordance was found in 32.3% (10/31) of concurrent infections. Regarding the identification of non-vaccine genotypes, the significantly prevalent genotypes in the anogenital area were HPV66 (12.6%, *p* < 0.0001), HPV53 (11.1%, *p* < 0.0001), HPV51 (8.7%, *p* < 0.0001), HPV42 (8.2%, *p* < 0.0001) and HPV68 (5.6%, *p* = 0.0034) in women and HPV66 (14.6%, *p* = 0.0058), HPV42 (12.2%, *p* = 0.0428), HPV51 (12.2%, *p* = 0.0428), HPV53 (12.2%, *p* = 0.0428), HPV70 (12.2%, *p* = 0.0428) and HPV73 (12.2%, *p* = 0.0428) in men. Considering the results of our study, we recommend including the high-risk genotypes HPV51, HPV68, HPV53 and HPV66 in future HPV vaccine formulations.

## 1. Introduction

The recent decades have witnessed major changes in the sexual behavior of both men and women, with the breaking of old taboos and rigid rules. There has been a lowering of the age of sexual debut, an increase in the number of sexual partners and general clearance of casual sex and formerly unacceptable sexual practices [1,2].

Recent statistical reports indicate how more than 80% of men and women engage in oral sex, making it the most common sexual practice along with vaginal sex [3,4,5,6]. Probably due to a lack of awareness of the risks and methods of protection, the use of condoms or dental dams during oral sex is minimal in both sexes, making oral transmission a foregone contagion route of sexually transmitted infections (STIs) [3,7,8].

Both viral and bacterial pathogens, such as human papillomavirus (HPV), Herpes simplex (HSV), Treponema pallidum and Chlamydia trachomatis, can be transmitted from the genital region to the oral cavity [7]. While an intact buccal and genital mucosa reduces the risk of acquiring such infections, the presence of abrasions, bleeding gums, gingivitis, periodontitis and ulcerations, even those caused by the STIs themselves, increase the chances of getting the infection [3,9]. 

As stated by World Health Organization (WHO), HPV is the most prevalent sexually transmitted pathogen, with approximately 80% of sexually active women and men being infected at least once in their lifetime [10,11]. 

Basal keratinocytes of mucosal and cutaneous epithelia are the targets of the infection, which is usually asymptomatic. Failure to eradicate the virus in a space of 12–24 months exposes the patient to the risk of developing benign and malignant hyperproliferative lesions [12]. In particular, epidemiological and biological data have allowed the distinction of mucosa-infecting types into high-risk HPVs (hrHPV) and low-risk HPVs (lrHPV), according to their degree of correlation with the development of malignant lesions [13,14]. 

It is estimated that more than 500,000 cases of cancer are caused by HPV each year, making it the most common virus associated with cancer in humans. The first neoplasm linked to hrHPVs was cervical cancer, of which almost all cases (99.5%) have an HPV-induced etiopathogenesis [10,15]. Later, other genital tumors have been associated with persistent HPV infection: 88% of anal cancers, 43% of vulvar, 70% of vaginal and 50% of penile cancers [10,16]. 

The recent rise of HPV-related head and neck cancers (HNSCC), specifically oropharyngeal squamous cell carcinoma (OPSCC) and oral squamous cell carcinomas (OSCC), has been linked to HPV transmission to the buccal mucosa [17,18,19]. In particular, oral HPV infection has been recognized as an independent risk factor for the development of OPSCC, which has been described in recent US-based statistics as the most common HPV-related tumors in men, with 17,222 cases registered from 2015 to 2019 [20,21].

On the other hand, lrHPV infections affecting the buccal mucosa can induce the development of benign oral papillary lesions, the most common of which are the following: condyloma acuminatum (CA), squamous papilloma (SP), verruca vulgaris (VV) and multifocal epithelial hyperplasia (MEH) [22]. Another manifestation of lrHPV infection is recurrent respiratory papillomatosis (RRP), affecting newborns and children infected by HPV-positive mothers through transplacental transmission or during vaginal delivery [23]. 

Nevertheless, oral HPV infections may be present even in the absence of visible oral mucosal abnormalities and may remain undetected for various periods [24]. 

Despite the well-known HPV tropism for both genital and oral sites and the rising concern about HPV-induced oral and oropharyngeal epithelial lesions, the exact route of transmission and the natural history of oral HPV infection are scarcely studied and still not fully understood [25].

Orogenital contact has been reported as the primary route of oral HPV transmission, with the likelihood of infection appearing to increase with the number of sexual partners, with continued exposure to risk determined by, for example, an HPV-positive partner and being in a homosexual relationship [23]. 

Some studies have suggested how autoinoculation, using the hands as intermediate contact and open-mouth kissing may also be a source of infection [23]. 

The conflicting results in the literature regarding the prevalence of oral HPV infection and its association with genital infection may be due to the diversity of oral sampling procedures and HPV detection methods, as well as the different interpretations of “concurrence” and “concordance” data [22,23,26].

Several studies have been conducted on possible strategies to combat HPV infection [27,28], however, the only currently valid route is vaccination.

To date, three different vaccines have been approved by the Food and Drug Administration (FDA) and consequently by the European Medicines Agency (EMA): the bivalent Cervarix, against HPV16 and 18, the quadrivalent Gardasil, against HPV6, 11, 16 and 18, and the nonavalent Gardasil 9, against HPV6, 11, 16, 18, 31, 33, 45, 52 and 58. HPV vaccination campaigns, generally targeting women, have been shown to be an effective prevention strategy for cervical and anal cancer [29,30,31,32].

Concerning oral infections, vaccine studies have shown encouraging but inconclusive results in preventing OPSCC and OSCC, with the search for evidence of efficacy complicated by the lack of specific premalignant lesions to use as surrogate endpoints [33,34,35]. 

Designing of HPV vaccines has been driven by the purpose of protecting against the genotypes most associated with cervical cancer, namely, HPV16 and HPV18, followed by HPV31, 33, 45, 52 and 58, and benign genital lesions, that is, HPV6 and 11. Recent studies describe how the same hrHPVs included in the vaccines cause 90–95% of HPV-related oropharyngeal cancers: an estimated 51–83% of HPV-positive patients had HPV16, while 0–20% had HPV31, 33, 45, 52 and 58, depending on the anatomical site [36,37]. 

However, changes in sexual practices, as described above, could lead to an increase in the prevalence of other genotypes in the future, both in the oral and in the genital region. This trend calls for the identification of other common genotypes not yet included in available vaccines so that their inclusion in future vaccines can eventually be considered.

In light of the above, in this observational retrospective study, we evaluate the prevalence, concurrence and concordance of oral and genital HPV in a cohort of unvaccinated male and female patients. Specifically, we defined concurrence as the simultaneous detection of one or more HPV types in genital and oral samples from the same patient, whereas hrHPV-concordance, lrHPV-concordance and type-concordance were defined as the detection of hrHPVs, lrHPVs and the same genotypes in both sites, respectively. 

The identification of high frequencies of HPV types not included in the quadrivalent and nonavalent vaccines at both sites could provide important clues about the possibility of expanding the genotypes included in future vaccine formulations.

## 2. Materials and Methods

### 2.1. Study Subjects and Samples Collection

The sample group consisted of 337 patients, 281 women and 56 men, referred for HPV testing to the Microbiology and Virology laboratory of the Department of Health Promotion, Mother and Child Care, Internal Medicine, and Medical Specialties (PROMISE) (Polyclinic “P. Giaccone” Hospital, University of Palermo). Patients underwent HPV testing following the detection of cervical and penile HPV-related lesions, or because they had a close contact with a positive person (e.g., a partner). Cervical scraping for women and urethral and genital scraping for men were performed by gynecologists and urologists in private practice.

To investigate whether the suspected genital infection had spread to the oral cavity, both female and male patients provided an oral rinse. Sampling was performed at the Oral Medicine with Dentistry for Fragile Patients Unit of the Polyclinic “P. Giaccone” Hospital, Palermo, by rinsing the oral cavity for 30 s with 10 mL of Original Mint Scope^®^ mouthwash (Procter & Gamble, Cincinnati, OH, USA), carefully reaching every part of the oral cavity and avoiding gargling. Before sampling, each patient was advised to avoid eating, drinking and using oral hygiene products. 

No funding or financial compensation was provided to study participants.

### 2.2. Samples Processing and HPV-DNA Extraction, Detection and Genotyping

To isolate the cellular component, each sample was centrifuged at 1600 rpm for 10 min. The pellet was resuspended in 1–4 mL of phosphate-buffered saline, PBS (EuroClone S.p.A, Milano, Italy) and then subjected to a second centrifugation at 13,000 rpm for 5 min. After discarding the supernatant, each pellet was stored at −20 °C or processed immediately.

A volume of 200 μL of oral pellets resuspended in PBS was used to extract DNA using the QIAamp Mini Kit (Qiagen, Hilden, Germany). HPV-DNA detection was carried out using INNOLiPA^®^ HPV Genotyping Extra II (Fujirebio, Tokyo, Japan), which identifies 23 genotypes classified as hrHPV or probable hrHPV (HPV16, 18, 31, 33, 35, 39, 45, 51, 52, 56, 58, 59, 67, 68, 26, 53, 66, 70, 73, 82) and 9 types classified as lrHPV (HPV6, 11, 40, 42, 43, 44, 54, 61, 62, 81, 83, 89). In this technique, a 65 bp fragment of the L1 gene, generated by an SPF10 PCR, is subjected to a reverse-dot blot hybridization assay. 

In samples that were HPV positive but not genotyped, a nested PCR was performed consisting of a first amplification with PGMY09/PGMY11 primers followed by a second amplification with GP05+/GP06+ primers as described elsewhere [38]. After direct Sanger sequencing of the amplicons, genotypes were obtained using Basic Local Alignment Search Tool (BLAST) sequence alignment.

### 2.3. Statistical Analysis 

Data are presented as numbers and percentages for categorical variables, while continuous data are expressed as the mean ± standard deviation (SD) unless otherwise specified. Particularly, medians with interquartile range (IQR) were used for variables with non-normal distribution.

The chi-square test and Fisher’s exact test were performed to evaluate significant differences in proportions or percentages between the two groups. Fisher’s exact test was used when the chi-square test was not appropriate. Chi-square goodness of fit was used to assess significant differences between three or more modalities of a variable. If the chi-square goodness of fit was significant (*p*-value < 0.05), residual analysis was performed using the Z-test. Test for normal distribution was performed using Shapiro–Wilk test. The *t*-test was used to test the differences between two means of unpaired data. Alternatively, the Mann–Whitney test was used when the distribution was not normal. Finally, all tests with *p*-value (*p*) < 0.05 were considered significant. Statistical analysis was performed using the Matrix Laboratory (MATLAB) analytical toolbox version 2008 (MathWorks, Natick, MA, USA) for Windows at 32 bits.

## 3. Results

The patient cohort consisted of 337 subjects, 83.4% (281) females and 16.6% (56) males, with a mean age of 33.7 (10.2) years (Table 1). 

HPV-DNA was detected in 73.6% (248/337) of genital samples, and 10.1% (34/337) of oral samples. Specifically, 73.7% (207/281) and 7.1% (20/281) of women were positive in the genital and oral sites, respectively, whereas men showed a genital positivity rate of 73.2% (41/56) and an oral positivity rate of 25% (14/56). Notably, no significant difference was observed between women and men at the genital site (*p* = 0.94), whereas a significant gender difference was observed at the oral site (*p* < 0.0001), i.e., we found a significant presence of HPV-positive men than HPV-positive women at the oral site.

Oncogenic genotypes, alone or together with lrHPVs, were detected in 86.7% (215/248) of cases in the genital area and 70.6% (24/34) in the oral cavity. Specifically, they were found in 87% (180/207) and 82.9% (34/41) of female and male genital specimens, respectively, and in 65% (13/20) and 78.6% (11/14) of female and male oral specimens, respectively.

Single infections were the most common in the oral cavity, with a rate of 91.2% (31/34), while single and multiple infections were detected in 62.1% (154/248) and 37.9% (94/248) of genital specimens, respectively (Table 1). For the sake of clarity, percentages referring specifically to women and men are shown in Table 1 rather than in the text.

Concurrence, namely, the simultaneous detection of one or more HPV types in genital and oral samples of the same patient, was found in 12.5% (31/248) of cases. In concurrent infections, hrHPV concordance, i.e., the detection of HR types in both anatomic sites, was detected in 61.3% (19/31) of patients, while lrHPV concordance was found in 12.9% of cases (4/31) (Table 1).

Type-concordance, i.e., the detection of the same HPV genotypes in both anatomic areas was found in 32.3% (10/31) of concurrent infection. 

Only three patients with a positive oral sample had a negative genital specimen, i.e., 3.4% (3/89), two men and one woman. 

The genotypes included in the quadrivalent and nonavalent vaccines were detected in 35.1% (87/248) and 61.7% (153/248) of genital samples and in 35.3% (12/34) and 44.1% (15/34) of oral samples (Table 1).

Figure 1 and Figure 2 show all the HPV genotypes detected in the patient’s cohort. Among the most common genotypes, those not included in the quadrivalent and nonavalent vaccines stand out (Table 2). In particular, the significantly most frequent genotypes not included in both vaccines were HPV66 (12.9%, 32/248, *p* < 0.0001), HPV53 (11.3%, 28/248, *p* < 0.0001), HPV51 (9.3%, 23/248, *p* = 0.0001), HPV42 (8.9%, 22/248, *p* = 0.0006) and HPV68 (6.0%, 15/248, *p* = 0.0081) in the genital area, while no significant differences were observed in the oral area for genotypes not included in the vaccines.

Table 3 shows the significant most frequent genotypes not included in both vaccines considering the sex. In particular, for the genital region in the women subgroup, we found HPV66 (12.6%, 26/207, *p* < 0.0001), HPV53 (11.1%, 23/207, *p* < 0.0001), HPV51 (8.7%, 18/207, *p* < 0.0001), HPV42 (8.2%, 17/207, *p* < 0.0001) and HPV68 (5.6%, 12/207, *p* = 0.0034); while in the men subgroup, we found HPV66 (14.6%, 6/41, *p* = 0.0058), HPV42 (12.2%, 5/41, *p* = 0.0428), HPV51 (12.2%, 5/41, *p* = 0.0428), HPV53 (12.2%, 5/41, *p* = 0.0428), HPV70 (12.2%, 5/41, *p* = 0.0428) and HPV73 (12.2%, 5/41, *p* = 0.0428). Even in this case, at the oral site, no significant differences were observed between genotypes not included in the vaccines in either the male or female subgroups.

Table 4 describes the comparison between HPV positivity and the variables reported in Table 1, i.e., age, sex and HPV risk classification at the two anatomical sites. The analysis considered the two groups of HPV-positive oral and HPV-positive genital samples. Patients with concurrent infection were included in both groups. 

No significant difference was found between the two groups for the age variable (median: 36 vs. 31, *p* = 0.09), while significant differences were observed for sex (male: 41.2% vs. 16.5%, *p* = 0.0007), i.e., considering only subjects positive for HPV, we found a significantly higher HPV positivity at the oral site than at the genital site in men (41.2% vs. 16.5%, *p* = 0.0007). As for HPV risk classification, we found a significantly higher frequency of patients with hrHPVs at the genital site than at the oral site (HR: 86.7% vs. 70.6%, *p* = 0.0143) (Table 4).

## 4. Discussion

The growth in unprotected oral sexual practices has made the transmission of STIs from the genital area to the oral mucosa a widespread event, which has been described for several pathogens. In particular, the transmission of HPV infection to the oral cavity is quite a hot topic, mainly because of the gradual increase in HPV-positive HNSCC observed in recent decades [39,40,41]. However, the natural history of oral HPV infection as well as the likelihood and the exact modes of transmission are not fully understood. At present, although HPV tropism is known for both genital and oral sites, there is still no consensus on the relationship between HPV infection in the genital tract and in the oral cavity [24,25,42]. 

HPV vaccines formulated on the basis of the most prevalent and disease-associated HPV genotypes are effective against genital HPV infection [29,30,31,32]. However, vaccines have shown promising but inconclusive results in preventing oral and oropharyngeal cancers, with proofs of efficacy complicated by the lack of specific premalignant lesions to use as surrogate endpoints [33,34,35].

Moreover, changes in sexual practices may result in the presence of other common genotypes, not yet included in the available vaccines, in both the oral cavity and in the genital area [2]. The identification of such genotypes in large cohorts of patients and in different geographical areas may provide an opportunity to improve future HPV vaccine formulations. 

The opportunity to analyze a cohort of men and women tested for oral and genital HPV in this study has allowed us to broaden our understanding of concurrent and concordant HPV infections, and, especially, to identify the most common HPV types that are not included in quadrivalent and nonavalent vaccines. This approach could represent a starting point for future vaccine design.

Coherently with what has been described in the literature, HPV-positive genital samples were more numerous than HPV-positive oral specimens (73.6% vs. 10.1%) [20]. 

Concerning oral infection, our findings fall within a wide range from 0% to 81% described in the literature. The lack of consistent data from one study to another may be explained by the diversity of oral sampling methods, HPV detection techniques and populations selected [43].

The percentage of genital HPV positivity was equally high in the women and man subgroups, while the oral HPV prevalence was significantly higher in men compared to women (25% vs. 7.1%, *p* < 0.0001). In addition to these findings, Table 4 shows the significantly higher rate of oral positive males compared to genital positive males. In other words, this figure suggests how men seem to have a greater tendency to develop oral than genital infection. 

These results are consistent with literature data describing men as having a higher prevalence of oral HPV infection, which is associated with an equally higher incidence of HPV-positive OPSCC [44]. It remains unclear why men and women are at different risks for oral HPV infection, especially since genital HPV infection is equally prevalent in both sexes. However, it has been suggested that a difference in immune response, notably weaker in men than in women, may contribute to a higher susceptibility to oral infection [44,45]. 

Some differences between oral and genital sites can be highlighted. First of all, although oncogenic genotypes were the most frequently detected in the two sites, in both women and men, their prevalence was significantly higher in genital samples than oral samples (86.7% vs. 70.6%, *p* = 0.0143). Additionally, oral samples were predominantly characterized by single infections (91.2% vs. 8.8%), unlike genital samples where multiple infections, although fewer than the single ones, accounted for a significant proportion of positive samples (37.9%).

Our data on concurrent orogenital infections show a different burden of oral infections between genital HPV-positive and HPV-negative patients. In fact, while 9.2% and 29.3% of women and men with a positive genital sample had an oral infection, only one woman and two men with no genital HPV infection had an HPV-positive oral sample. These results are consistent with what has been observed in several other studies [20,46,47,48,49,50]. 

Furthermore, these figures regarding orogenital HPV concordance show the difference between women and men (9.2% vs. 29.3%) that, even in this case, has already been reported in the literature [46]. In terms of type-specific concordance, we add our findings to literature reports that suggest a percentage of around 30% for both sexes [47,49,51].

Although these results suggest an association between genital infection and the onset of oral infection, it should be noted that any consideration about concurrence and concordance must take into account the small numbers of the groups analyzed.

The focus on non-vaccine HPVs detected in both areas allowed the high prevalence of certain genotypes to be identified. Specifically, the hrHPV genotypes HPV51, HPV68, HPV53 and HPV66 and the lrHPV genotype HPV42 were the most frequently detected genotypes not included in the commercially available vaccines in the genital samples of the whole cohort. A closer look at the two groups of female and male patients allowed the same genotypes to be identified as significant in the female samples; in males, however, statistical significance was lost for HPV68, while HPV70 and HPV73 were found to be significantly represented.

Other studies carried out in different geographical areas, mainly in China, have shown similar results. HPV51 has been reported in several studies to be one of the most common circulating genotypes in women, similar to HPV53 and HPV68 [52,53,54,55,56]. As for HPV66, Zheng et al. also identified it among the significantly prevalent hrHPV in Chinese women [57].

Looking at Italian data, Serretiello et al. recently identified HPV51 as one of the most common genotypes and HPV42 as the prevalent lrHPV in women [58]. Kiwerska et al. also highlight a high prevalence of HPV42 in both women and men of their cohort, both in oral and in genital swabs [59].

The lack of studies evaluating genital HPV prevalence exclusively in men does not allow a proper comparison with our results. 

Unfortunately, the same evaluation performed on oral samples did not reveal a significantly high prevalence of non-vaccine types.

The classification of HPV as carcinogenic, probably carcinogenic or possibly carcinogenic by the International Agency for Research on Cancer (IARC) is based on the identification of strong epidemiological, biological and/or biochemical evidence. Focusing on the non-vaccine genotypes identified as significantly represented in this study, HPV51 is classified as “carcinogenic”, HPV68 as “probably carcinogenic”, while HPV53 and 66 are defined as “possibly carcinogenic to humans”. HPV51 has been shown to cause a small percentage, which varies greatly from region to region, of cervical cancers, but it has the ability to interfere with cellular tumor suppression mechanisms. The same can be said for HPV53 and HPV66, which are common in the general population but rarely found in cervical cancer samples. HPV73 has also been defined as suggestive of carcinogenicity [13].

Our observations are limited by the small number of patients, especially males. The disproportion between male and female groups was due to the way patients accessed the Oral Medicine Unit with Dentistry for Fragile Patients. As oral specimen collection often followed genital examination, women undergoing regular gynecological examination and cervical HPV testing formed the majority of our study group.

Nevertheless, our results, although preliminary, have potential public health implications. In fact, the identification of common non-vaccine genotypes, which have also been reported in other studies and whose carcinogenic potential is also described by IARC data, suggests the next steps in HPV vaccination strategies. Based on our results, it would be advisable to consider including the high-risk genotypes HPV51, HPV53 and HPV66 in future vaccine formulations, as they are significantly prevalent in both female and male genital samples, and to a lesser extent HPV68, due to its high prevalence in women.

## Figures and Tables

**Figure 1 vaccines-11-01466-f001:**
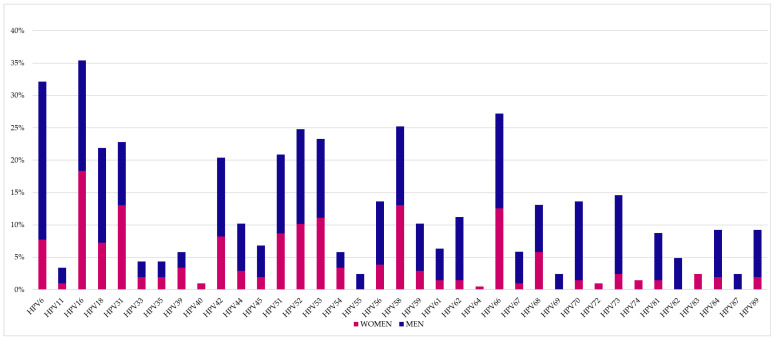
Percentages of genotypes detected at the genital site.

**Figure 2 vaccines-11-01466-f002:**
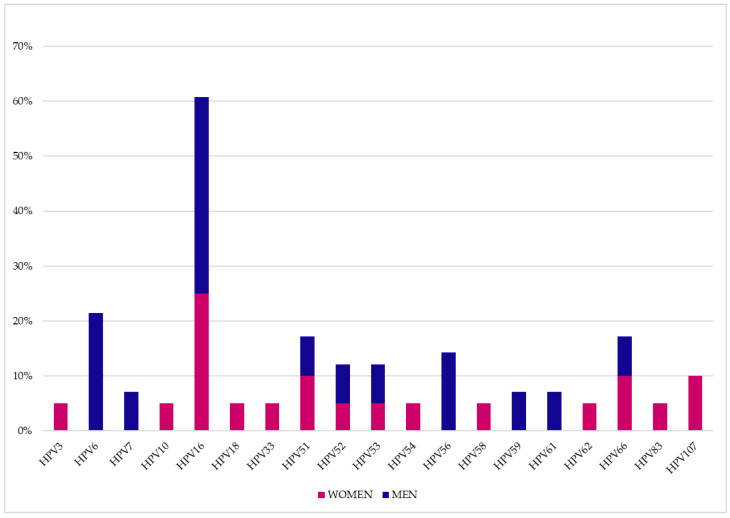
Percentages of genotypes detected at the oral site.

**Table 1 vaccines-11-01466-t001:** General characteristics of the cohort.

Parameters	Total (*n* = 337)	Females (*n* = 281)	Males (*n* = 56)
Age			
Mean (SD)	33.7 (10.2)	32.4 (9.2)	38.2 (11.9)
Median (IQR)	31 (25.9–40)	30.3 (25.5–37.1)	38.2(29.7–45.4)
HPV+			
Genital	73.6% (248/337)	73.7% (207/281)	73.2 (41/56)
Oral	10.1% (34/337)	7.1% (20/281)	25% (14/56)
HrHPV (including lrHPV/hrHPV infections)			
Genital	86.7% (215/248)	87% (180/207)	82.9% (34/41)
Oral	70.6% (24/34)	65% (13/20)	78.6% (11/14)
Single infections			
Genital	62.1% (154/248)	67.1% (139/207)	36.6% (15/41)
Oral	91.2% (31/34)	95% (19/20)	85.7% (12/14)
Multiple infections			
Genital	37.9% (94/248)	32.9% (68/207)	63.4% (26/41)
Oral	8.8% (3/34)	5% (1/20)	14.3% (2/14)
Concurrence	12.5% (31/248)	9.2% (19/207)	29.3% (12/41)
hr-concordance	61.3% (19/31)	57.9% (11/19)	66.7% (8/12)
lr-concordance	12.9% (4/31)	15.8% (3/19)	8.3% (1/12)
Type-concordance	25.8% (10/31)	31.6% (6/19)	33.3% (4/12)
Genotype included in the quadrivalent vaccine			
Genital	35.1% (87/248)	31.9% (66/207)	51.2% (21/41)
Oral	35.3% (12/34)	30% (6/20)	42.9% (6/14)
Genotype included in the nonavalent vaccine			
Genital	61.7% (153/248)	59.4% (123/207)	73.2% (30/41)
Oral	44.1% (15/34)	40% (8/20)	50% (7/14)

lr = low risk; hr = high risk.

**Table 2 vaccines-11-01466-t002:** Genotype not included in quadrivalent and nonavalent vaccines at oral and genital sites. Frequencies are indicated in brackets. Significant and most frequent genotypes in groups are marked with an asterisk.

	HPV+ (Oral) (*n* = 34)		HPV+ (Genital) (*n* = 248)	
Genotypes Not Included in Quadrivalent or Nonavalent Vaccine	% (*n*)	L/H Risk	% (*n*)	L/H Risk
3	2.9 (1)	LR	-	
7	2.9 (1)	LR	-	
10	2.9 (1)	LR	-	
35	-	-	2.0 (5)	HR
39	-	-	3.2 (8)	HR
40	-	-	0.8 (2)	LR
42	-	-	8.9 (22) *	LR
44	-	-	3.6 (9)	LR
51	8.8 (3)	HR	9.3 (23) *	HR
53	5.9 (2)	HR	11.3 (28) *	HR
54	2.9 (1)	LR	3.2 (8)	LR
55	-	-	0.4 (1)	LR
56	5.9 (2)	HR	4.8 (12)	HR
59	2.9 (1)	HR	3.6 (9)	HR
61	2.9 (1)	LR	2.0 (5)	LR
62	2.9 (1)	LR	2.8 (7)	LR
64	-	-	0.4 (1)	HR
66	8.8 (3)	HR	12.9 (32) *	HR
67	-	-	1.6 (4)	HR
68	-	-	6.0 (15) *	HR
69	-	-	0.4 (1)	HR
70	-	-	3.2 (8)	HR
72	-	-	0.8 (2)	LR
73	-	-	4.0 (10)	HR
74	-	-	1.2 (3)	LR
81	-	-	2.4 (6)	LR
82	-	-	0.8 (2)	HR
83	2.9 (1)	LR	2.0 (5)	LR
84	-	-	2.8 (7)	LR
87	-	-	0.4 (1)	LR
89	-	-	2.8 (7)	LR
107	5.9 (2)	LR	-	-

**Table 3 vaccines-11-01466-t003:** Genotype not included in quadrivalent and nonavalent vaccines and present at both oral and genital sites considering the gender. Frequencies are indicated in brackets. Significant and most frequent genotypes in groups are marked with an asterisk.

	HPV+ (Oral) (*n* = 34)		HPV+ (Genital) (*n* = 248)	
Genotypes Not Included in Quadrivalent or Nonvalent Vaccine	Males (14) % (*n*)	Females (*n* = 20) % (*n*)	Males (*n* = 41) % (*n*)	Females (*n* = 207) % (*n*)
3	-	5.0 (1)	-	-
7	7.1 (1)	-	-	-
10	-	5.0 (1)	-	-
35	-	-	2.4 (1)	1.9 (4)
39	-	-	2.4 (1)	3.4 (7)
40	-	-	-	1.0 (2)
42	-	-	12.2 (5) *	8.2 (17) *
44	-	-	7.3 (3)	2.9 (6)
51	7.1 (1)	10.0 (2)	12.2 (5) *	8.7 (18) *
53	7.1 (1)	5.0 (1)	12.2 (5) *	11.1 (23) *
54	-	5.0 (1)	2.4 (1)	3.4 (7)
55	-	-	2.4 (1)	-
56	14.3 (2)	-	9.8 (4)	3.9 (8)
59	7.1 (1)	-	7.3 (3)	2.9 (6)
61	7.1 (1)	-	4.9 (2)	1.4 (3)
62	-	5.0 (1)	9.8 (4)	1.4 (3)
64	-	-	-	0.5 (1)
66	7.1 (1)	10.0 (2)	14.6 (6) *	12.6 (26) *
67	-	-	4.9 (2)	1.0 (2)
68	-	-	7.3 (3)	5.6 (12) *
69	-	-	2.4 (1)	-
70	-	-	12.2 (5) *	1.4 (3)
72	-	-	-	1.0 (2)
73	-	-	12.2 (5) *	2.4 (5)
74	-	-	-	1.4 (3)
81	-	-	7.3 (3)	1.4 (3)
82	-	-	4.9 (2)	-
83	-	5.0 (1)	-	2.4 (5)
84	-	-	7.3 (3)	1.9 (4)
87	-	-	2.4 (1)	-
89	-	-	7.3 (3)	1.9 (4)
107	-	10.0 (2)	-	-

**Table 4 vaccines-11-01466-t004:** Characteristics and comparison between oral and genital HPV-positive patients and age, gender and HPV risk classification variables.

Parameters	HPV+ (Oral) (*n* = 34)	HPV+ (Genital) (*n* = 248)	HPV+ (Oral) vs. HPV+ (Genital)
Age			
Mean (SD)	37.1 (11.7)	33.6 (10.2)	
Median (IQR)	36 (28.5, 46.5)	31 (25.75, 40)	0.09 (MW)
Shapiro–Wilk test	*p* = 0.0025, rN	*p* < 0.0001, rN	
Gender			
Male	41.2% (14)	16.5% (41)	0.0007 * (C)
Female	58.8% (20)	85.5% (207)	
hrHPV (including lrHPV/hrHPV infections)	70.6% (24)	86.7% (215)	
lrHPV	29.4% (10)	13.3% (33)	0.0143 * (C)

* = significant test; C = chi-square test; hr = high risk; lr = low risk; MW = Mann–Whitney test.

## Data Availability

Not applicable.

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
