# Peer review of "Orogenital Human Papillomavirus Infection and Vaccines: A Survey of High- and Low-Risk Genotypes Not Included in Vaccines"

_vaccines, 2023, doi:10.3390/vaccines11091466_

Round 1

Reviewer 1 Report

No further comments

Author Response

Thank you very much for your positive report and for the time that you have taken to write it.

Reviewer 2 Report

This manuscript is interesting and meaningful,however, it dose not write whether the patients were vaccinated against HPV,as we know that vaccination has impact on infected HPV genotype. Therefore,please add HPV vaccination history of enrolled patients, and to compare the HPV genotype between vaccinated and unvaccinated patients.

The analysis of results is not detailed enough,for example,the HPV typing comparing in two sites of same patients is important,but not written in the article. In addition,based on Fig.1 and Fig.2, the positive rate of some HPV types varies greatly between man and women,why?please explain.

Author Response

[Reviewer]: This manuscript is interesting and meaningful, however, it dose not write whether the patients were vaccinated against HPV,as we know that vaccination has impact on infected HPV genotype. Therefore,please add HPV vaccination history of enrolled patients, and to compare the HPV genotype between vaccinated and unvaccinated patients.

[Reply]: Thank you for your comment. All patients were unvaccinated, as the age (mean: 33.7 years) was too high to allow adherence to the Italian vaccination programme. To make this point clearer we decided to include this information in the text.

[Reviewer]: The analysis of results is not detailed enough,for example,the HPV typing comparing in two sites of same patients is important,but not written in the article.

[Reply]: Thank you for your comment. The focus of our paper was the identification of the most frequent genotypes not included in the vaccines at the two sites, and not a possible comparison of genotypes found at the two sites on the same HPV+ patients. Hence, we believed that a comparative analysis between genotypes included or not included in one site versus another, would not be useful for our study, unless that genotype is more frequent than the other genotypes in both groups or single group, for the purposes of its inclusion in the vaccine.

The only comparison between the groups (Tab 4) was to see whether there were differences between the two groups (oral and genital) in terms of age, gender and the type of genotype found for a better description of the data. Particularly, the analysis shows a non-significant difference for age, but a significant difference for gender, showing a greater presence of positive males in the oral group (41.2% vs. 16.5%, p=0.0007) and high-risk genotypes more present in subjects in the genital group (86.7% vs. 70.6%, p=0.0143) as commented in the Discussion section.

[Reviewer]: In addition,based on Fig.1 and Fig.2, the positive rate of some HPV types varies greatly between man and women,why?please explain.

[Reply]: The difference in the prevalence of some HPV types between men and women is an interesting result, which would benefit further analysis. As far as we know, this issue is not addressed in the literature, but a possible explanation could be a different susceptibility to certain genotypes caused by the specific features of sex-dependent differences in the immune response. The same explanation is reported in the text (Page 10, lines 297-300) regarding the different risks of acquiring oral infection between men and women. Another reason could be the different characteristics of the female and male genitalia, such as the greater mucosal surface in women compared to men.

Reviewer 3 Report

This is a state of the art retrospective study evaluating orogenital HPV concurrence, high risk HPV, low risk HPV as well as subtype concordance in a samples of 337 persons. The authors also estimate the prevalence of the most common genotypes in order to suggest modifications in HPV vaccines.

The manuscript is very interesting, well written and appropriate for the journal. A few comments:

 1.       In the statistical analysis is mentioned: “while continuous data are expressed as the mean ± standard deviation (SD) unless otherwise specified”. In general you may report median and quartiles 1 and 3, under the condition that normality does not hold, please clarify in this section when mean and SD are not reported.

2.       Figures 1 & 2 in my opinion should be combined in a single figure with the subtypes aligned to be the same subtype of the genital system under the oral, thus the reader will be able to see the differences immediately.

3.       In table 4 the MW: Mann Whitney is not mentioned.

Author Response

[Reviewer]:  In the statistical analysis is mentioned: “while continuous data are expressed as the mean ± standard deviation (SD) unless otherwise specified”. In general you may report median and quartiles 1 and 3, under the condition that normality does not hold, please clarify in this section when mean and SD are not reported.

[Reply]: Thank you for your suggestion. We added in the statistical analysis section the sentence: Particularly, medians with Interquartile Range (IQR) were used for variables with non-normal distribution.

[Reviewer]: Figures 1 & 2 in my opinion should be combined in a single figure with the subtypes aligned to be the same subtype of the genital system under the oral, thus the reader will be able to see the differences immediately.

[Reply]: We have tried to follow your suggestion, but combining data on oral and genital infections in male and female patients results in a very confusing graph. We have therefore decided to keep two separate graphs.

[Reviewer]: In table 4 the MW: Mann Whitney is not mentioned.

[Reply]: Thank you for your suggestion. We added it.

Reviewer 4 Report

It would be pertinent to include information regarding the different methods of obtaining oral samples for HPV detection. The oral rinse may be compared with an oropharyngeal swab.

It may also be necessary to state if the subjects were heterosexual, homosexual/lesbian or bisexual. 

Would these HOV also be cleared as how it happens with HPV in cervix?

Have the HPV types described herein been isolated from oropharyngeal tumours?

This is important for any recommendations on vaccination.

Author Response

[Reviewer]: It would be pertinent to include information regarding the different methods of obtaining oral samples for HPV detection. The oral rinse may be compared with an oropharyngeal swab.

[Relpy]: Thank you for the suggestion. In our laboratory, rinsing is used as the only collection method for HPV testing, as it is described in the literature as the best method for collecting oral exfoliated cells.

[Reviewer]: It may also be necessary to state if the subjects were heterosexual, homosexual/lesbian or bisexual.

[Relpy]: Thank you for your comment, It would have been a piece of interesting information to include in our paper, but as we decided to focus on other patients' characteristics, this information was not collected. In the future, when enrolling new patients, we will certainly have care in gathering this information.

[Reviewer]: Would these HOV also be cleared as how it happens with HPV in cervix?

[Relpy]: As described in the review by Wierzbicka et al. (2022), although the acquisition of persistent, and therefore oncogenic, oral HPV infection is rarer than genital infection, it appears that in healthy individuals, infections at both sites tend to clear at approximately the same rate.

[Reviewer]: Have the HPV types described herein been isolated from oropharyngeal tumours? This is important for any recommendations on vaccination.

[Relpy]: All the patients were subjected to oral sampling after the identification of a suspected genital HPV infection, so they did not have any clinical suspicion of oral ailments